# Toward Semantic Gaze Target Detection

**Samy Tafasca**
Idiap Research Institute
École Polytechnique Fédérale de Lausanne
stafasca@idiap.ch

**Anshul Gupta**
Idiap Research Institute
École Polytechnique Fédérale de Lausanne
agupta@idiap.ch

**Victor Bros**
Idiap Research Institute
École Polytechnique Fédérale de Lausanne
vbros@idiap.ch

**Jean-Marc Odobez**
Idiap Research Institute
École Polytechnique Fédérale de Lausanne
odobez@idiap.ch

## Abstract

From the onset of infanthood, humans naturally develop the ability to closely observe and interpret the visual gaze of others. This skill, known as gaze following, holds significance in developmental theory as it enables us to grasp another person's mental state, emotions, intentions, and more [6]. In computer vision, gaze following is defined as the prediction of the pixel coordinates where a person in the image is focusing their attention. Existing methods in this research area have predominantly centered on pinpointing the gaze target by predicting a gaze heatmap or gaze point. However, a notable drawback of this approach is its limited practical value in gaze applications, as mere localization may not fully capture our primary interest — understanding the underlying semantics, such as the nature of the gaze target, rather than just its 2D pixel location. To address this gap, we extend the gaze following task, and introduce a novel architecture that simultaneously predicts the localization and semantic label of the gaze target. We devise a pseudo-annotation pipeline for the GazeFollow dataset, propose a new benchmark, develop an experimental protocol and design a suitable baseline for comparison. Our method sets a new state-of-the-art on the main GazeFollow benchmark for localization and achieves competitive results in the recognition task on both datasets compared to the baseline, with $40\%$ fewer parameters.

## 1 Introduction

Gaze is an important marker in non-verbal communication that is indicative of a person's visual attention. It is also a proxy measure of cognition and can be used to evaluate a subject's intentions, preferences, emotions, among others. Consequently, it received a lot of attention over the years from different research communities such as neuroscience [13], psychology [17], cognitive science [44], robotics [43], and education [37].

In computer vision, the analysis and understanding of attention was formulated through different tasks. One research direction focuses on predicting a gaze direction representing the 3D line of sight from a frontal image of a face [34]. Another one tries to estimate the visual focus of attention (VFOA), *i.e.* the gaze target of a person, given 3D information about the subject (*e.g.* body, head, eyes) and the scene (*e.g.* layout, object positions) [45, 4]. Beyond predicting gaze as a standalone signal, several research efforts focused on understanding gaze dynamics in the context of social communication such as inferring mutual gaze [35, 15, 20], joint attention [14, 15, 20], or gaze aversion [15].

This paper focuses on the task of gaze following [42], which extends the idea of Visual Focus of Attention (VFOA). Gaze following aims to predict the 2D pixel coordinates where a person in an

38th Conference on Neural Information Processing Systems (NeurIPS 2024).

image is looking. The major benefit of this formulation is that it makes no assumptions about the scene and doesn't require additional equipment, such as wearable devices. However, a notable drawback is that solely predicting the pixel location of the gaze target often falls short for real-world applications that demand additional information, such as object class or social gaze class.

One possibility to address this limitation is to post-process the output of a gaze following method by verifying if a gaze point falls within the bounding box of a detected object [10]. However, this multi-stage process entails additional computation, often requiring the use of additional pre-trained models, leading to inefficiency and less than optimal results. Furthermore, pre-trained detectors typically ignore uncountable objects (*e.g.* wall, sea) which are often possible gaze targets. Finally, unlike object detection, in gaze following we predict heatmaps and not boxes, which makes applying a separate object detector afterwards challenging. In such case, how do we match a gaze heatmap to the right object box? We could consider the gaze point (*i.e.* $\arg\max$) as mentioned before, but what if the point falls within multiple boxes? And what if the heatmap is multimodal, and the $\arg\max$ happens to land on the wrong target? The joint training of the gaze heatmap and the gaze target class, aside from being the more natural formulation, allows the model to learn the best way to dynamically make sense of the heatmap in order to infer the right class.

An alternative approach is to frame the problem as a Human-Object-Interaction (HOI) recognition task, where *looking* serves as an interactive action. However, many existing HOI datasets lack consistent and systematic labeling of the looking behavior. To the best of our knowledge, V-COCO [22] is the only dataset doing so, but it is limited to 80 object classes, which may not encompass the full spectrum of potential gaze targets found in images, including cases where individuals look at locations categorized as *stuff* semantic classes. Moreover, other datasets focus on a limited number of classes (*e.g.*, watching TV or a cell phone), thereby biasing the looking task towards specific objects. Consequently, utilizing an HOI verb-object task formulation with existing HOI datasets might result in learning spurious correlations, wherein the detection of a particular object (*e.g.* *TV*) strongly suggests a specific verb (*e.g.* *watching*). This is a well known issue in HOI and has fostered the development of benchmark datasets specifically designed to evaluate HOI methods based on evidence rather than relying on dataset-specific correlation priors [25]. Given these limitations, there is a clear need to explore the task in a novel manner and develop new datasets and protocols to address this research topic effectively.

In this paper, we propose an end-to-end architecture that predicts both the localization and the class label of the gaze target, addressing the limitations above. Furthermore, we frame the recognition part as a visual-text alignment task, which offers the benefit of generalizing to other classes beyond the training vocabulary. To this end, we make the following contributions

- We address, for the first time, the semantic gaze following problem by devising a visual-language architecture that efficiently tackles both localization and target class categorization tasks simultaneously.
- We introduce novel benchmarks, a new baseline for comparison, and experimental protocols for investigating the extended task, drawing from datasets within the gaze following and HOI communities.
- Our architecture sets a new state-of-the-art in gaze target localization on the main GazeFollow benchmark dataset, and demonstrates strong categorization performance compared to more complex and computationally intensive baselines.

## 2  Related Work

**Gaze Target Detection.** Traditional methods for estimating the Visual Focus of Attention (VFOA) [47, 1, 39, 18, 43, 2, 36] were limited by their reliance on specific scene or activity priors, which hindered their generalization to more arbitrary settings where such priors could not be provided. To overcome these limitations, Recasens et al. [41] proposed a novel formulation of the problem, aiming to infer the 2D image coordinates that correspond to the scene target being observed by a person in the image. Standard methods for gaze following adopt a two-branch architecture, comprising a scene branch for saliency detection and a head analysis branch for the person of interest to infer a gaze direction. Information from the two branches is then fused to predict the final gaze heatmap. This type of architecture has demonstrated robust performance in gaze following, with the ability to also predict if the person is looking within the image or outside [9, 32, 48, 49, 26, 16, 5, 27].

**Semantic Gaze Following.** Building on this foundation, we introduce an architecture that not only estimates gaze target localization but also identifies the class label of the gaze target. To the best

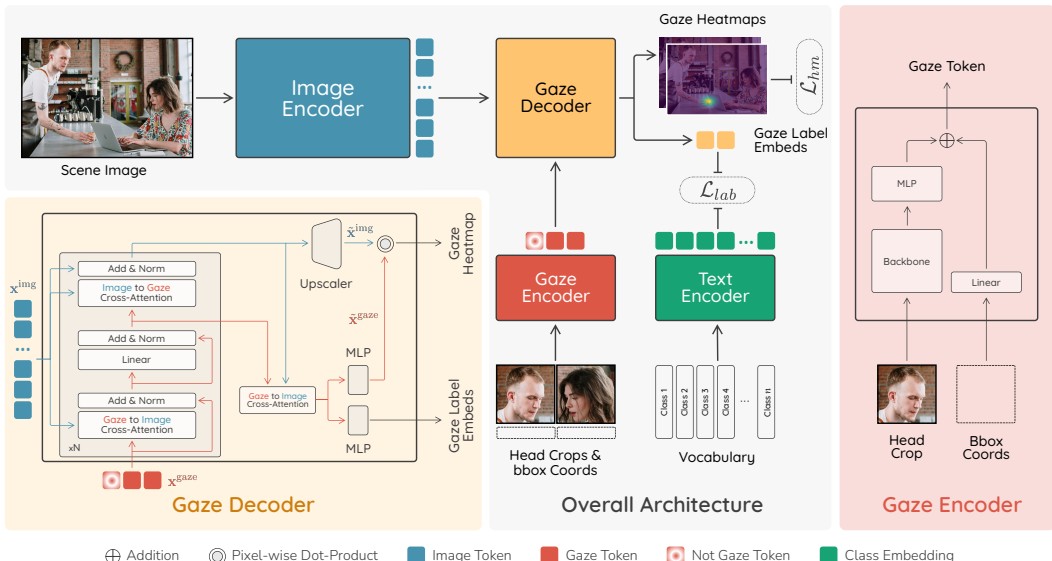

Figure 1: Overview of our architecture. [A] The scene image is passed through an Image Encoder to produce image tokens (blue squares). [B] The head crops and head box coordinates are processed by a Gaze Encoder to generate gaze tokens (orange squares). [C] The image and gaze tokens are fed to a Gaze Decoder to predict both the gaze heatmaps and (normalized) gaze label embeddings (yellow squares) through a series of cross-attention operations. [D] The text encoder computes (normalized) class embeddings (green squares) based on a predefined vocabulary of concept classes. [E] Finally, we compute similarity scores between the predicted gaze label embeddings and vocabulary embeddings.

of our knowledge, we are the first to address this task. The closest work in this direction is from Tonini et al. [50] which proposed a model that simultaneously predicts the gaze heatmap and detects objects in the scene. Their approach involves a primary branch that identifies both the head and object bounding boxes within a scene, followed by feature extraction and gaze cone estimation to predict the best candidate object of focus in the cone and its corresponding heatmap. It is important to note that object detection here only serves as an auxiliary task to improve gaze localization performance and is not meant to produce a class label. In fact, the target object is often not detected, and only a bounded vocabulary set is supported. Our work seeks to address this limitation by adopting a weakly supervised contrastive objective aiming to align vision and text.

## 3 Architecture

Inspired by the idea of promptable segmentation [30], we design an architecture (See Figure 1) for promptable gaze following where the scene image is processed separately from people. We decode gaze outputs by *prompting* the encoded image representation using person-specific information (*i.e.* head crop and box coordinates). This is achieved by a lightweight transformer decoder that can process multiple people at the same time or separately in a batch. The rationale behind this design is for the expensive scene encoding operation to be performed only once in a person-agnostic manner. Then, after we obtain the gaze tokens of people, their gaze targets (*i.e.* location and label) can be decoded efficiently. Intuitively, we expect the image representation to identify gaze target candidates (*i.e.* salient objects), and for the decoding step to act as a filtering mechanism that selects the right gaze target based on the query person. We empirically verify this hypothesis in the qualitative analysis section. It is important to note that this design is in stark contrast to most previous two-stream gaze following approaches [10, 16, 27, 21] that fuse the person and image representations early on. This makes them inefficient for multi-person inference because the expensive encoding is repeated for each individual. We provide more details about each component below.

### 3.1 Image Encoder

Given an input image $\mathbf{I} \in \mathbb{R}^{H \times W \times C}$, we employ a transformer encoder to produce image tokens $\mathbf{x}^{\text{img}} \in \mathbb{R}^{N \times D}$, where $N = w \cdot h$ is the number of patches, $w = \frac{W}{P}$, $h = \frac{H}{P}$, $P$ is the patch size, and $D$ is the dimension of the transformer. We can also use a convolutional backbone as the image encoder, assuming we equip the output feature map with positional information.

## 3.2 Gaze Encoder

The goal of the gaze encoder is to process the head crop of the target person at a higher resolution (*e.g.* $224 \times 224$), as well as its location in the image in order to produce a position-sensitive embedding capturing directional gaze information. Given an input head crop $\mathbf{h}_{\text{crop}} \in \mathbb{R}^{H \times W \times C}$, we pass it through a backbone followed by a 2-layer MLP to output a gaze token $\mathbf{x}^{\text{gaze}} \in \mathbb{R}^D$. Additionally, we project the corresponding head bounding box $\mathbf{h}_{\text{bbox}} \in \mathbb{R}^4$ to $D$ using a linear layer and add it to the gaze token to make it aware of the person's position in the image. If we process $N_p$ persons, we get the final gaze tokens $\mathbf{x}^{\text{gaze}} \in \mathbb{R}^{N_p \times D}$. Please note that we can batch process multiple people in different images by merging the image and person dimensions as the batch dimension. However, this requires the use of padding or truncation of people to ensure $N_p$ is the same across all images.

## 3.3 Gaze Decoder

The goal of the gaze decoder is to combine information from the scene and the people in order to predict the gaze heatmaps and label embeddings. It is composed of a transformer with $N$ blocks, an upscaler module and two MLPs for the final predictions.

Once the image is encoded, the image tokens $\mathbf{x}^{\text{img}}$ and gaze tokens $\mathbf{x}^{\text{gaze}}$ go through the transformer decoder blocks, which are composed of two cross-attention operations and a feed-forward module, with residual connections and normalization layers in-between. The cross-attention operations go in two ways: one where the gaze tokens $\mathbf{x}^{\text{gaze}}$ generate the queries, while the image tokens $\mathbf{x}^{\text{img}}$ generate the keys and values (we call this $\mathbf{x}^{\text{gaze}} \rightarrow \mathbf{x}^{\text{img}}$ or gaze-to-image cross-attention), and one where image tokens $\mathbf{x}^{\text{img}}$ generate the queries while the gaze tokens $\mathbf{x}^{\text{gaze}}$ generate the keys and values (we call this $\mathbf{x}^{\text{img}} \rightarrow \mathbf{x}^{\text{gaze}}$ or image-to-gaze cross-attention). This two-way design allows image tokens to incorporate information from gaze tokens, and vice-versa. This helps them better align for the final dot-product operation that will produce the predicted gaze heatmap.

After the $N$ blocks, we transform the gaze tokens one last time using a final $\mathbf{x}^{\text{gaze}} \rightarrow \mathbf{x}^{\text{img}}$ cross-attention layer. The output is an updated version of the gaze and image tokens, which we also call $\mathbf{x}^{\text{gaze}} \in \mathbb{R}^{N_p \times D}$ and $\mathbf{x}^{\text{img}} \in \mathbb{R}^{N \times D}$ to ease notation. In Figure 1, $\mathbf{x}^{\text{img}}$ flows through the decoder following the blue arrows, while $\mathbf{x}^{\text{gaze}}$ flows through the orange arrows.

To obtain the final predictions, we use the task-specific MLPs in the following manner: (i) the gaze tokens $\mathbf{x}^{\text{gaze}} \in \mathbb{R}^{N_p \times D}$ are passed through a Gaze Label MLP composed of 6 layers to predict the gaze label embeddings $\mathbf{g}_{\text{lab}} \in \mathbb{R}^{N_p \times D_l}$, where $D_l$ is the dimension of the output gaze label embedding, (ii) the gaze tokens $\mathbf{x}^{\text{gaze}}$ are fed to a Gaze Heatmap MLP composed of 3 layers to project them to a lower dimensional embedding $\tilde{\mathbf{x}}^{\text{gaze}} \in \mathbb{R}^{N_p \times d}$, where $d < D$. At the same time, the image tokens are rearranged into a spatial feature map $\mathbf{x}^{\text{img}} \in \mathbb{R}^{h \times w \times D}$ and upscaled to the resolution of the output heatmap while reducing the number of channels $\tilde{\mathbf{x}}^{\text{img}} \in \mathbb{R}^{H_{\text{hm}} \times W_{\text{hm}} \times d}$. The Upscaler module is composed of two blocks of one interpolation followed by one convolutional layer each. Finally, for each projected gaze token, we apply a dot-product with every spatial position of the upscaled image tokens to obtain the final heatmaps $\mathcal{H} \in \mathbb{R}^{N_p \times H_{\text{hm}} \times W_{\text{hm}}}$. Essentially, this dot-product acts as a filtering mechanism that localizes the gaze target of the subject thanks to the alignment done by the decoder between the gaze token and the image tokens at the right spatial positions.

Additionally, the gaze decoder uses a learnable token called the *not gaze token* whose role is to expand the range of the $\mathbf{x}^{\text{img}} \rightarrow \mathbf{x}^{\text{gaze}}$ cross-attention. Recall that this operation updates the image representation with information from the gaze tokens *i.e.* each image token gets updated with a weighted sum of the (values of the) gaze tokens. Conceptually, the *not gaze token* allows tokens of image regions where nobody focuses their attention to be updated by a different "null" token instead of forcing a weighted sum on the existing gaze tokens. We verify this hypothesis in a later section.

## 3.4 Text Encoder

Since we frame the prediction of the gaze label as a text-vision alignment task, our architecture also requires a text encoder to convert the vocabulary of class labels into text embeddings. We chose CLIP [40] as our text encoder, and we kept it frozen.

During training, the predicted gaze label embedding is aligned with the available class label embeddings. During inference, we first build a vocabulary of class labels, pass them through the text encoder to get their embeddings, then compare the predicted gaze label embedding with each one of them and select the class label with the highest similarity score.

### 3.5 Losses

**Heatmap Loss ($\mathcal{L}_{reg}$).** It is the standard pixel-wise MSE between the GT and the predicted heatmaps:

$$\mathcal{L}_{hm} = \sum_{x,y}^{W_{hm}, H_{hm}} ||\mathcal{H}_{x,y}^{\text{gt}} - \mathcal{H}_{x,y}^{\text{pred}}||_2^2$$

**Label Loss ($\mathcal{L}_{lab}$).** The label loss follows a similar formulation to the multimodal contrastive InfoNCE used in [40] with two main differences: 1) our loss is not symmetric (*i.e.* we only consider the image to text loss component), and 2) since our language labels are (pseudo-) classes and not full captions, there will inevitably be class redundancies within each batch (*i.e.* two or more samples having the same gaze label), so we consider the unique ground-truth gaze labels in each batch. Formally, given a batch of $N$ predicted visual gaze label embeddings $I_i = \mathbf{g}_{\text{lab},i}^{pred}$ and their associated ground-truth class embeddings $T_i = \mathbf{g}_{\text{lab},i}^{gt}$ and ground-truth classes $y_i$, the loss can be expressed as:

$$\mathcal{L}_{lab} = -\frac{1}{N} \sum_{i=1}^{N} \log \left( \frac{\exp\left(s(I_i, T_i)/\tau\right)}{\sum_{j \in \mathcal{U}} \exp\left(s(I_i, T_j)/\tau\right)} \right)$$

where $\tau$ is a learnable temperature parameter, $s(a, b)$ is the similarity between $a$ and $b$, and $\mathcal{U}$ is a minimum subset of indices in $[\![1, N]\!]$ which identifies all classes in the batch, *i.e.* $\{y_i, i \in \mathcal{U}\} = \{y_i, i = 1 \ldots N\}$ and $\forall(j, j') \in \mathcal{U}^2, y_j \neq y_{j'}$. This loss aims to maximize the cosine similarity between each predicted visual gaze embedding and the corresponding class embedding, while minimizing the similarity between the negative pairs. This formulation bears a resemblance to the idea of supervised contrastive learning presented in [29].

**Angular Loss ($\mathcal{L}_{ang}$).** Optionally, we can append a second MLP head to the backbone in the gaze encoder in order to predict a normalized gaze direction vector from the input head crop. This vector can be supervised using the angular loss, which is defined based on the cosine of the angle between the predicted and ground truth gaze vectors according to:

$$\mathcal{L}_{ang} = 1 - <\mathbf{g}_{\text{vec}}^{gt}, \mathbf{g}_{\text{vec}}^{pred}>$$

where $<a, b>$ denotes the inner product between $a$ and $b$. While the anguler loss doesn't influence the final performance, we found it to be very useful in interpreting model predictions. For example, it helps to understand whether a failure mode is due to the head processing part (*e.g.* when the face is not visible), or the target selection part. Furthermore, it can be informative in real-world applications when the gaze heatmap is not reliable (*e.g.* when the person is looking outside the frame).

**Global loss.** The global loss is a given by: $\mathcal{L} = \lambda_{hm}\mathcal{L}_{hm} + \lambda_{lab}\mathcal{L}_{lab} + \lambda_{ang}\mathcal{L}_{ang}$

## 4 Datasets

### 4.1 GazeFollow

In order to train our end-to-end architecture, we need annotations for the semantic label of the gaze target. Unfortunately, there is no gaze following benchmark that provides annotations for both the gaze target position and class label. In order to solve this problem, we design a pseudo-annotation pipeline to automatically infer the semantic label of the ground-truth gaze target. Considering that gaze following datasets come with point annotations for gaze targets, our approach is to first segment the image, then match the gaze point with the predicted semantic class of the underlying pixel. Since we need to segment the images entirely, it is important to ensure that the segmentation we perform incorporates an open vocabulary that is able to describe any object encountered in the dataset.

To this end, we use the GazeFollow dataset [42], and leverage two open-source projects that implement open-world segmentation using various foundational vision and language models. The first method, known as RAM-Grounded-SAM[1], generates precise masks and accurate semantic labels. However, it has a tendency to overlook many regions in the image. To fill this gap and pseudo-annotate the missed gaze instances, we utilize a second method called Semantic-Segment-Anything[2]. This tool provides more comprehensive coverage of the image, but it comes with the trade-off of introducing noisy labels and oversegmentation — where many small segments are constituents of a larger object. We provide a comparison of the two segmentation methods in the supplementary.

---

[1] https://github.com/IDEA-Research/Grounded-Segment-Anything
[2] https://github.com/fudan-zvg/Semantic-Segment-Anything

Running this pipeline on our dataset produces $\sim 5000$ unique labels. We perform a series of text processing operations to clean these labels (*i.e.* correct incomplete words, remove stopwords, duplicates, adjectives, and prepositions), eventually leading to a reduced vocabulary of $\sim 3700$ pseudo-labels.

Finally, in order to reliably test our models, we manually annotate the test set of GazeFollow with target classes. This annotation process is not restricted to a predefined set of objects, but we ensure that the labels are consistent (*e.g.* by avoiding synonyms when possible). Also, since the area where a person is looking, represented as a heatmap, often includes multiple objects, we also annotated other possible gaze targets whenever possible. For example, a person cutting a cake is probably looking at both the cake and the knife. This multi-label annotation approach also helps to deal with the ambiguity related to object hierarchy (*e.g.* a person looking at the *wheel* of a car is also looking at the *car*, so it makes sense to assign both to the gaze instance). At the end, we obtained a test set ground-truth vocabulary of 346 classes (*cf.* the supplementary for a word cloud).

## 4.2 GazeHOI

Aside from GazeFollow, we also introduce a new benchmark for the simultaneous localization and recognition of the gaze target. Since the assignment of a class label is predicated on determining the location of the gaze target first, it is very difficult to manually annotate a new dataset from scratch. This is because we can not define a vocabulary of object classes that we want to annotate in advance, and ensure people are only looking at those objects.

To circumvent this problem, we repurpose existing human-object interaction datasets (*i.e.* annotations for person box, object box, object class and interaction verb) to create GazeHOI. The process is described as follows: (i) we combine 5 HOI datasets (V-COCO [22], HICO-DET [7], HCVRD [53], SWiG-HOI [51], HOI-A [33]), (ii) for each dataset, we manually select a subset of verbs from the top 200 frequent ones where the person is also likely looking at the object (*e.g. cut* or *repair*, but not *carry*), (iii) we use an off-the-shelf head detector to detect head bounding boxes and match them to the annotated person's bounding box, then we filter out instances without detected heads, (iv) for each head-object HOI instance, we draw their bounding boxes on the corresponding image, (v) a team of annotators looks at these images, and answers *yes* or *no* based on whether or not the bounding boxes are correct, and the person is looking at the object they are interacting with, and (vi) discard *no* instances. This process solves our problem, and has the benefit of reducing the usually expensive manual annotation to a simpler manual verification.

At the end, we obtain a total of 43808 images and 58146 instances after discarding about 50% of images in the verification step. Each instance is annotated with the person's body and head boxes, the object box, the object class, and the interaction verb. The vocabulary has 985 object classes, from which we isolate 522 rare classes (*i.e.* less than 10 instances each) into a separate split. This can be used for future research on open-vocabulary semantic gaze following, similar to the practice adopted in open-vocabulary object detection [19]. The final dataset used in our experiments features a vocabulary of 463 object labels and 55995 gaze instances which we split into 47214/3781/5000 for the train, val and test sets. Last but not least, we also run a deduplication pipeline to ensure that the validation and test sets contain no images from the training sets of GazeHOI or GazeFollow since both of them are based on popular overlapping vision benchmarks. In Figure 2, we show a few samples from GazeHOI (*cf.* the supplementary for a word cloud of the vocabulary).

# 5 Experiments

## 5.1 Baseline

Aside from comparing with previous gaze target localization methods from the literature, we also propose a strong baseline to assess our gaze target recognition performance given that no such work was attempted before. To this end, we design a 2-stage method as follows: first, we freeze our proposed gaze model and use it to predict a gaze heatmap. Next, we use this heatmap to condition the original image by emphasizing the focused area. Then, we apply the CLIP's [40] pretrained vision model on the resulting image to produce the gaze label embedding. The idea is to leverage the alignment between CLIP's vision and text encoders. Finally, in terms of conditioning the input image based on the predicted heatmap, we consider three variants: masking, blurring and cropping. Figure 3 shows an overview of the baseline as well as the different conditioning variants.

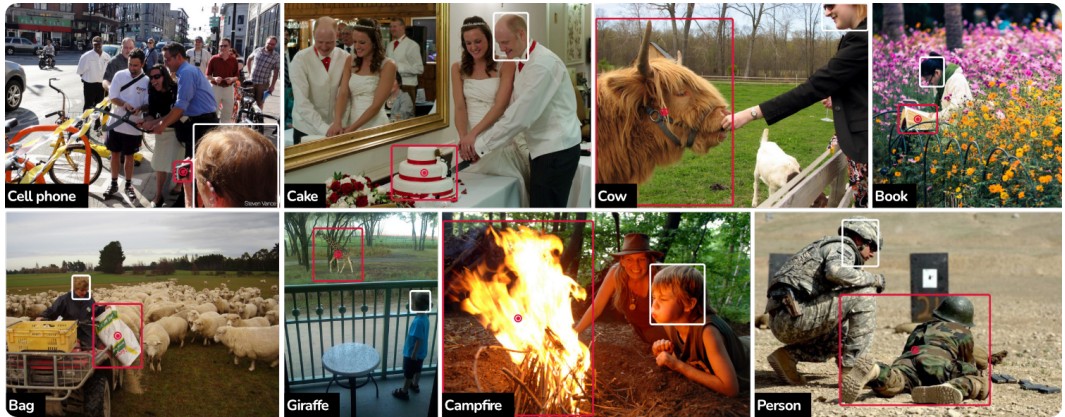

Figure 2: Samples from GazeHOI. We show the head box (white) and the object's box (red).

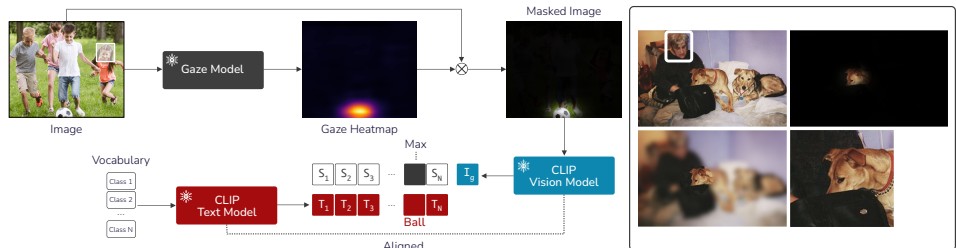

Figure 3: [Left] Overview of the proposed baseline architecture using masked conditioning. The similarity scores $S_i$ are computed from the gaze label embedding $I_g$ and the class embeddings $T_i$. [Right] Comparison of the conditioning variants based on the predicted gaze heatmap: original image (top left), masking (top right), blurring (bottom left), and cropping (bottom right).

As an additional benefit, since our model's predicted label is dependent on the predicted gaze location, using our own gaze model to generate the heatmap for the baseline allows us to control for the localization factor, enabling an unbiased comparison of recognition performance.

## 5.2 Comparison with the State-of-the-art

We summarize our quantitative results on the GazeFollow benchmark in Table 1 (the experimental protocol is provided in the supplementary). In terms of localization performance, our architecture sets a new state-of-the-art across both metrics, surpassing the second best method of Tafasca *et al.* by $4.4\%$ and third best of Jin *et al.* by $8.5\%$ on the Avg. Dist metric. In terms of recognition, our method outperforms all 3 variants of the frozen baseline by a significant margin, *i.e.* more than $20\%$ flat accuracy points compared to the second best (*i.e.* crop variant).

Since CLIP has probably not been trained on many images that are heavily masked, blurred or cropped, we decided to add two more variants by fine-tuning CLIP's vision encoder: (1) we directly fine-tune the baseline (crop) on GazeFollow. During training, the cropping is based on the ground-truth heatmap instead of a predicted one. (2) instead of feeding CLIP modified images, we use the original ones and apply the conditioning at the feature level. Specifically, we perform a weighted average of the output image tokens (*i.e.* from the CLIP vision encoder applied to the input image) based on the (downscaled) heatmap (*i.e.* predicted by the gaze model), followed by a projection. Since the heatmap might not cover the entire object, this last variant allows the model to have access to the surrounding context. The performance of these two variants is much closer to our method, where the cropping one is slightly worse, and the heatmap weighting is slightly better.

It is important to remember two key facts about the design of our baselines: (i) They build upon our model's localization ability (*i.e.* the most accurate to date) which contributes to recognition performance. To verify this, we replaced the gaze heatmap prediction part in the best version of the baseline by the model from [10], and found that performance drops and becomes worse than our proposed model (refer to Table 1, Baseline†). (ii) They add an entire vision transformer that

| Method | Localization | | Recognition | | |
|---|---|---|---|---|---|
| | Avg. Dist. ↓ | Min. Dist. ↓ | Acc@1 ↑ | Acc@3 ↑ | MultiAcc@1 ↑ |
| Random | 0.471 | 0.391 | 0.002 | 0.010 | 0.003 |
| Bias / Majority | 0.295 | 0.229 | 0.010 | 0.015 | 0.011 |
| Chong *et al.* [10] | 0.137 | 0.077 | — | — | — |
| Fang *et al.* [16] | 0.124 | 0.067 | — | — | — |
| Jin *et al.* [27] | 0.118 | 0.063 | — | — | — |
| Bao *et al.* [5] | 0.122 | — | — | — | — |
| Tafasca *et al.* [48] | 0.125 | 0.064 | — | — | — |
| Jin *et al.* [26] | 0.126 | 0.076 | — | — | — |
| Tafasca *et al.* [49] | 0.113 | 0.057 | — | — | — |
| Baseline (Mask) | 0.108 | 0.051 | 0.124 | 0.253 | 0.147 |
| Baseline (Blur) | 0.108 | 0.051 | 0.190 | 0.362 | 0.222 |
| Baseline (Crop) | 0.108 | 0.051 | 0.239 | 0.428 | 0.278 |
| Baseline (Crop) fine-tuned | 0.108 | 0.051 | 0.437 | 0.588 | 0.504 |
| Baseline (heatmap weight) | 0.108 | 0.051 | **0.466** | **0.653** | **0.542** |
| Baseline[†] (heatmap weight) | 0.137 | 0.077 | 0.442 | 0.620 | 0.514 |
| Ours | **0.108** | **0.051** | 0.447 | 0.642 | 0.516 |

Table 1: GazeFollow dataset. The best scores are given in bold, while the second best are underlined. All baselines use our own model for gaze heatmap prediction, except for Baseline[†] which uses [10].

is pretrained on a large-scale dataset specifically designed for such semantic recognition tasks. In terms of parameter count, our model features 116M while the baseline has 200M. This is because our model only uses an MLP head (3.3M parameters) for label prediction while the baseline uses a ViT (86.6M parameters) to encode the semantics of the image separately. That is a decrease of 42% in the total number of parameters (*cf*. the supplementary material for a comparison of FLOPS). We later show in our ablations that performance drops significantly if we try to use the same CLIP vision encoder to do both localization and recognition, which emphasizes the need for the baseline to have two specialized ViT encoders. Finally, upon manual inspection of test set samples where our model and the best performing heatmap-weighted baseline don't agree, we noticed that most of them are failures due to hierarchy or semantic similarity (*e.g.* prediction of *ball* then *racket* when the ground-truth is *racket*). Please refer to the supplementary material for some qualitative examples to illustrate this comparison.

| Method | Localization | Recognition | |
|---|---|---|---|
| | GazeAcc ↑ | Acc@1 ↑ | Acc@3 ↑ |
| Random | 0.166 | 0.002 | 0.006 |
| Bias / Majority | 0.352 | 0.082 | 0.083 |
| Baseline (Mask) | 0.723 | 0.197 | 0.298 |
| Baseline (Blur) | 0.723 | 0.306 | 0.458 |
| Baseline (Crop) | 0.723 | 0.388 | 0.546 |
| Baseline (Crop FT) | 0.723 | 0.617 | 0.707 |
| Baseline (Hm weight) | 0.723 | **0.646** | **0.748** |
| Ours[†] | 0.652 | 0.306 | 0.463 |
| Ours | **0.723** | 0.583 | 0.706 |

Table 2: Results of our model and baselines on the Gaze-HOI dataset. The best scores are given in bold, while the second best are underlined. The † sign means the model was trained on GazeFollow and evaluated on GazeHOI without fine-tuning.

On GazeHOI (*cf*. Table 2), we observe a similar trend with our method outperforming the zero-shot baselines by a large margin, while both fine-tuned variants outperform our model on recognition scores. We note that the pre-trained GazeFollow model is already able to perform the localization task (*i.e.* 65% *vs.* 72% for Gaze Accuracy). However, the zero-shot performance is much lower than the fine-tuned counterpart (*i.e.* 30% *vs.* 58%). This is probably due to the mismatch between the pseudo-labels seen in GazeFollow and the vocabulary of Gaze-HOI (*i.e.* about 150 classes are new).

Finally, we need to emphasize that unlike the baselines, our gaze label prediction head is trained from scratch, so we believe that the small scale of the datasets also plays an important role in limiting recognition performance (*i.e.* $100K$ *vs.* $400M$ instances).

## 5.3 Model Analysis & Ablations

**Image Encoder.** We hypothesized previously that our architecture design should steer the image encoder into capturing possible gaze target candidates. In order to verify this claim, we visualize in the second row of Figure 4 the attention maps from the last layer of the image encoder. It is very clear that the model is highlighting specific regions that correspond to salient objects (*e.g.* heads, hands, documents, tables, signboard, *etc.*).

**Batch Size.** Contrastive learning approaches are known to benefit from larger batch sizes [40, 8]. We vary our batch size from 48 to 300 and find that performance plateaus at the top end (*cf.* Table 3).

| Experiment | Avg. D. ↓ | Acc@1 ↑ | Acc@3 ↑ |
|---|---|---|---|
| $BS = 72$ | 0.109 | 0.405 | 0.601 |
| $BS = 144$ | 0.109 | 0.437 | 0.623 |
| $BS = 216$ | 0.108 | 0.450 | 0.636 |
| $BS = 300$ | 0.108 | 0.447 | 0.642 |
| $\mathcal{L}_{hm}$ | 0.107 | – | – |
| $\mathcal{L}_{lab}$ | – | 0.417 | 0.604 |
| $\mathcal{L}_{hm}/\mathcal{L}_{lab}$ | 0.108 | 0.451 | 0.640 |
| $\mathcal{L}_{hm}/\mathcal{L}_{lab}/\mathcal{L}_{ang}$ | 0.108 | 0.447 | 0.642 |
| $N_p = 5$ | 0.113 | 0.437 | 0.619 |
| $N_p = 3$ | 0.112 | 0.433 | 0.620 |
| $N_p = 1$ | 0.108 | 0.450 | 0.636 |
| bbox | 0.132 | 0.353 | 0.562 |
| head crop | 0.133 | 0.357 | 0.565 |
| bbox & head crop | 0.107 | 0.386 | 0.592 |

Table 3: Ablation results on GazeFollow for batch size, losses, number of people during training ($BS = 216$), and gaze encoder ($BS = 72$ and resolution of $448 \times 448$).

**Losses.** We also perform an ablation to assess the impact of the different losses. Based on Table 3, we note that training to recognize the gaze target's label without explicit localization downgrades Acc@1 performance by a flat $3.4\%$. On the other hand, adding the label loss does not seem to influence localization performance. However, even if it is not reflected on the current datasets and metrics, we believe that supervising using semantic information could improve localization since the task of selecting what a person looks at in a given context, after a field of view has been determined, is inherently a semantic problem.

**Number of People.** We note that increasing the number of people during training progressively degrades performance (*cf.* Table 3). Given our late-fusion approach, a higher $N_p$ puts more pressure on the decoder to accommodate multiple gaze tokens. Then, the image tokens need to be updated in a way that selectively aligns with all gaze tokens simultaneously at different spatial positions.

**Gaze Encoder.** To verify whether our elaborate gaze encoder is necessary, we experiment with an encoder that only processes the box coordinates, and another one that only takes the head crop without location. To compensate for the high resolution of head crops, we conduct this ablation with an image resolution of $448 \times 448$ and a batch size of 72 because of hardware constraints. In the first case, the burden shifts towards the image encoder which now has to capture salient objects, and also understand people and their gaze so that querying with the head position retrieves the answer immediately. On the other hand, training with only the head crop will probably force the model to infer the head location by matching head features to image features. Based on Table 3, we see that both versions degrade localization performance significantly.

**Not Gaze Token.** Quantitatively speaking, this extra token seems to improve performance, especially for the localization metrics (*i.e.* Avg Dist of $0.112$ *vs.* $0.115$ for models trained with $N_p = 3$ and batch size of $216$). To better understand what's happening, we visualize the $\mathbf{x}^{\text{img}} \rightarrow \mathbf{x}^{\text{gaze}}$ cross-attention map from the last decoder block. Since the input is a single person, each image token in the cross-attention operation is updated based on a weighted average of the gaze token and the *not gaze token*. The last row of Figure 4 shows the weight value $w$ of the gaze token (meaning that the weight of the learnable token is $1 - w$). The red areas indicates image tokens that gave more importance to the gaze token, and vice-versa for the blue. It is clear from the samples that the decoder is using the *not gaze token* as a placeholder for the background, or any area not being looked at by the input person, allowing for a better discrimination of gaze-relevant image regions.

## 6 Discussion

**Limitation and Future Work.** While our model has shown good performance in recognizing the gaze target class, the current design is not equipped to deal with the multi-label nature of the problem. Specifically, if multiple objects are highlighted by the gaze heatmap, it would be desirable for the model to predict them all (*e.g. knife*, *oignon*, and *chopping board*). Since the model is limited to

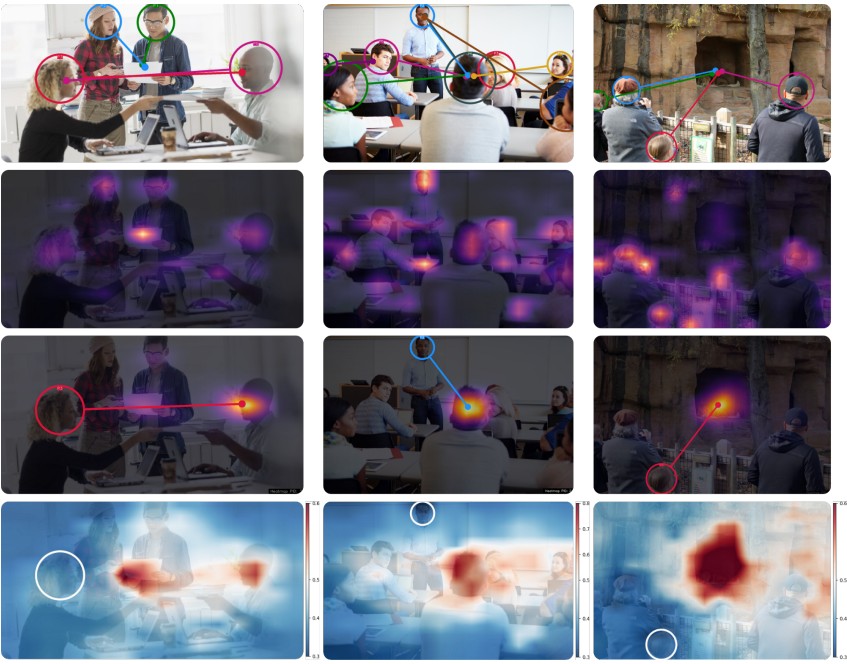

Figure 4: Qualitative samples from our model on images from the internet. The top row shows gaze point predictions for all people. The second row show the last attention map from the image encoder. The third row shows a predicted heatmap of a single person. The last row shows the weight of the gaze tokens in the last image to gaze cross-attention of the decoder for the same person.

predicting one gaze label embedding, we think it will likely produce some weighted average of those objects' embeddings. Furthermore, while the GazeFollow dataset is relatively small, there are significantly more objects in the images than annotated gaze instances. These additional semantic pseudo-labels (from the segmentation) can also be leveraged through auxiliary losses to align spatially localized image content with their textual counterpart (a similar approach was used in [31]) in order to enhance recognition performance. We leave the investigation of these ideas to future work.

**Societal Impact.** Gaze following methods can bring tremendous value in many real-world applications that foster positive change in society (*e.g.* screening neurodevelopmental disorders). However, care must be taken when deploying this technology in order to avoid privacy violations, and mitigate risks of malfunction in sensitive applications (*e.g.* surveillance systems). We encourage the community to use these models responsibly.

# 7   Conclusion

In conclusion, our study represents a step forward in gaze target detection by extending the traditional gaze following formulation to incorporate the class label of the target. Our proposed architecture has successfully integrated semantic understanding into the task while maintaining state-of-the-art performance in terms of localization. To this end, we leveraged a weakly supervised training regime based on pertinent pseudo-labels derived from open-world segmentation pipelines. Naturally, we employed a contrastive learning objective to align the visual embedding representing the gaze target with its textual counterpart, thereby allowing for some flexibility to build specialized vocabularies during inference based on the application. We hope that our code, datasets, model checkpoints and research insights will pave the way for future research on semantic gaze following.

**Acknowledgement.** This research has been supported by the AI4Autism project (Digital Phenotyping of Autism Spectrum Disorders in Children, grant agreement number CRSII5 202235/1) of the the Sinergia interdisciplinary program of the SNSF.

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

# A  Supplementary Material

## A.1  Qualitative Evaluation

We show multiple qualitative samples on the GazeHOI dataset, both correct predictions and failure cases in Figures 5, and 6 respectively.

Overall, common classes like *book*, *cellphone*, *bicycle* are usually well localized and correctly identified. It's interesting to see that the model performs well in spite of the diversity of the samples in terms of scenes, people, activities, interactions, distance from the camera, viewing angle and lighting conditions.

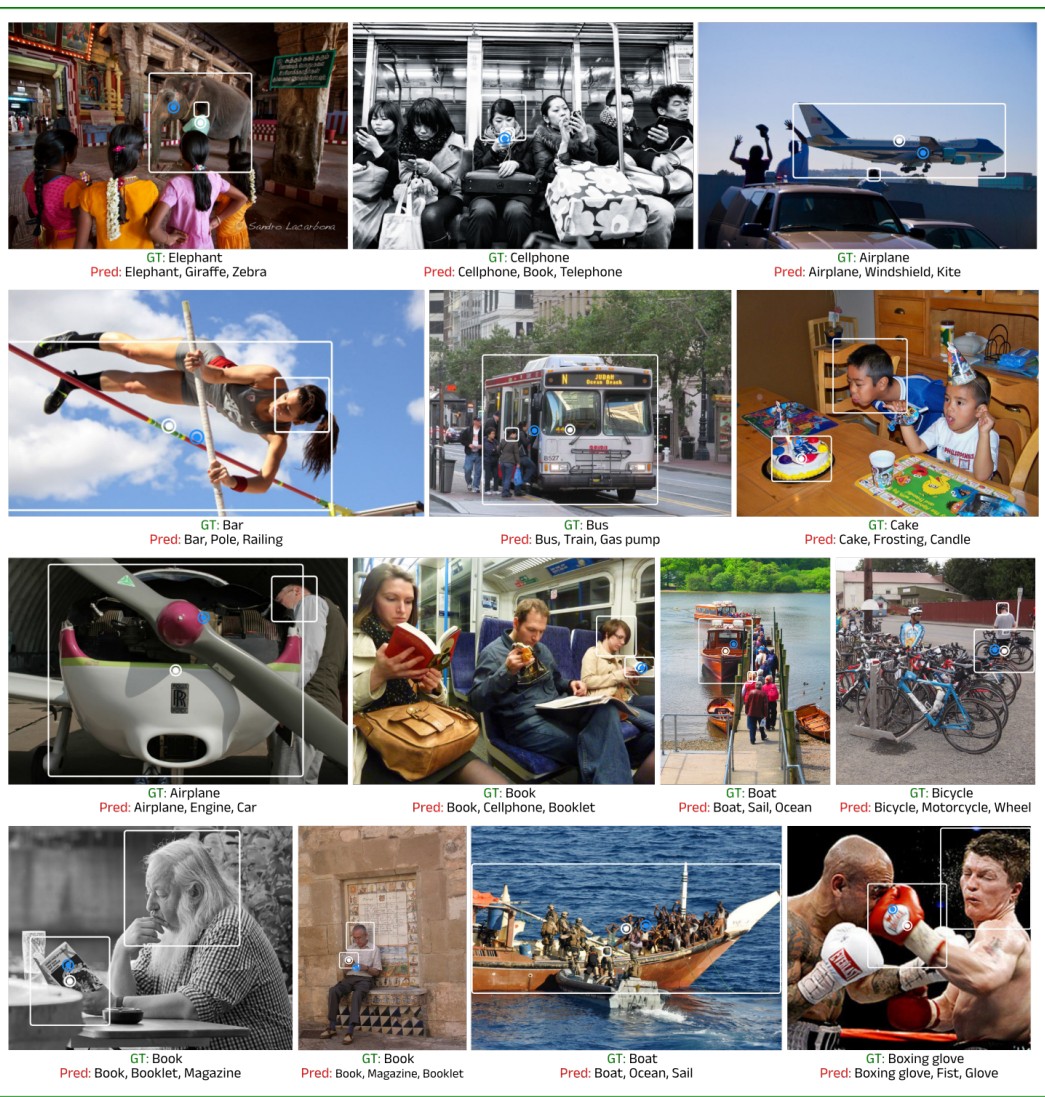

Figure 5: Samples with correct predictions on GazeHOI. The bounding boxes in white represent the ground-truth, the predicted gaze point is shown in blue. We also show the 3 top predicted class labels, and the corresponding ground-truth label.

Looking at the failure cases, some are simply difficult because the object is barely visible (*e.g.* the *bag* in the second image of the last row). Even if it's correctly localized, if there is occlusion, or the object is too small, the model will naturally tend to predict the class of the closest larger object that it knows. In this case, it tries to predict the animal *deer*, but since it's not in the vocabulary, it retrieves *giraffe* and *zebra* instead. In other cases, the target label is correct when the location is off. This can happen when the image features multiple instances of the same object class, and the model localizes the wrong one (*e.g.* *bird* in second image of the third row). Finally, some failure cases are simply due

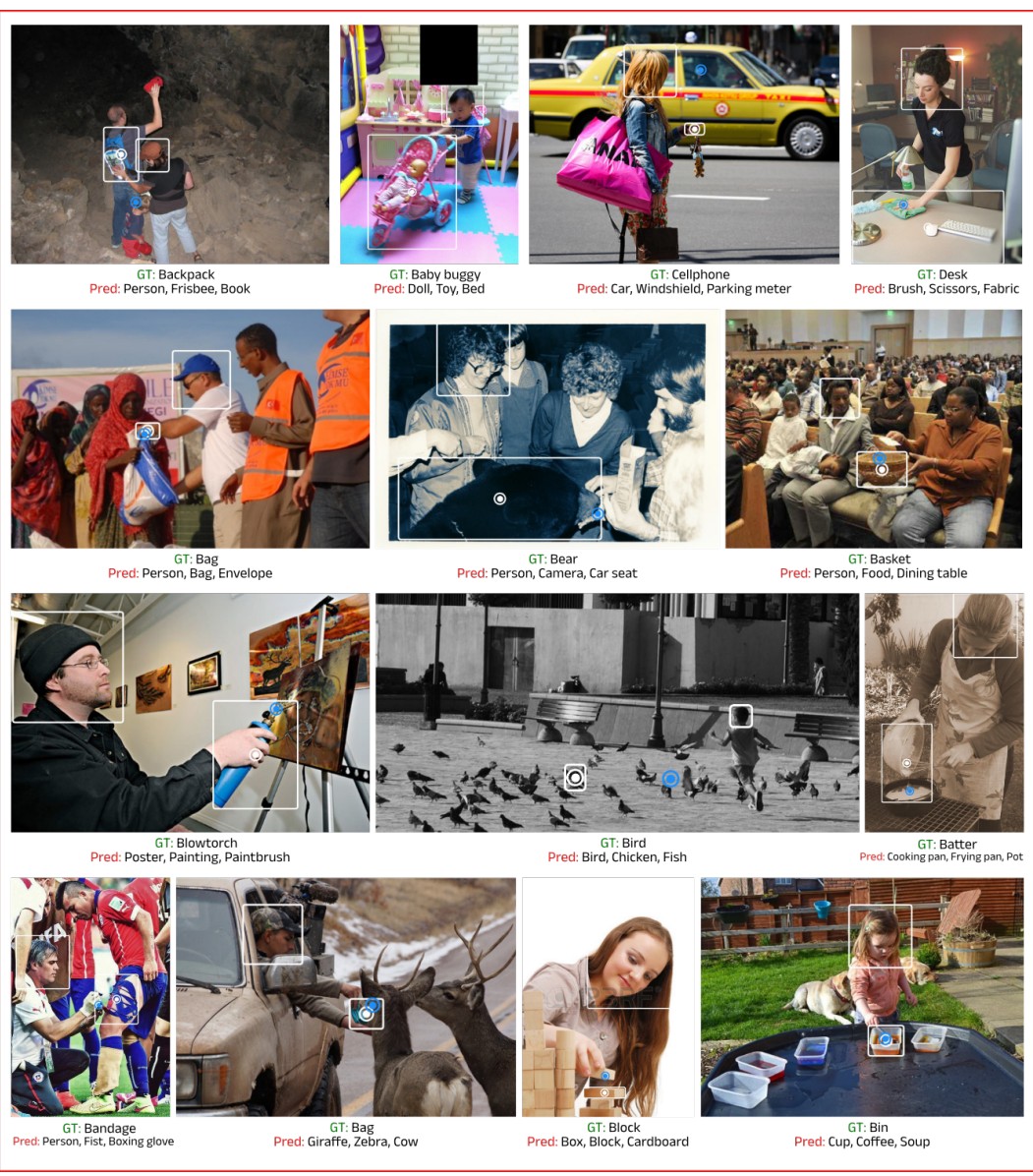

Figure 6: Samples with incorrect predictions on GazeHOI (*i.e.* either the class label or the gaze point is not matching the ground-truth). The bounding boxes in white represent the ground-truth, the predicted gaze point is shown in blue. We also show the 3 top predicted class labels, and the corresponding ground-truth label.

to the multi-label nature of the problem. For example, in the first image of the third row, the model predicts *poster* and *painting*, while the ground-truth is *blowtorch*. Technically both the *painting* and the *blowtorch* are in the vicinity of the person. We observe similar patterns on qualitative samples from the GazeFollow dataset in Figure 7.

## A.2   Baseline Comparisons & Limitations

Due to the localization task being framed as a heatmap prediction, our model's design is well-equipped to dynamically infer the correct target label. This means that it will not necessarily output the object class corresponding to the area of the highest intensity in the heatmap when the heatmap is multimodal. The baseline however, pushes the class label to match the main peak of the heatmap by design. In Figure 8 (top), we illustrate this behavior using different samples where the predicted heatmap is bimodal, with the right target corresponding to the smaller peak. In this case, our model

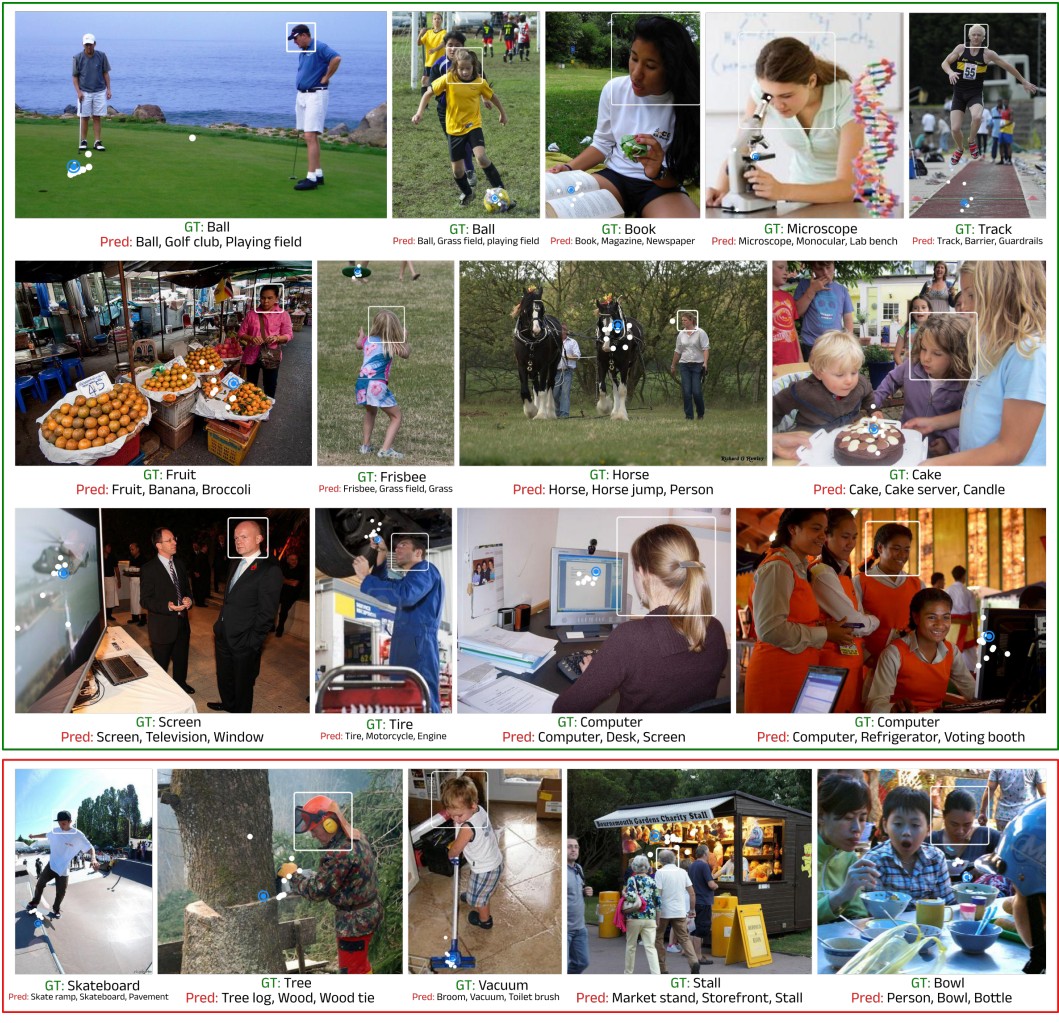

GT: Ball
Pred: Ball, Golf club, Playing field
GT: Ball
Pred: Ball, Grass field, playing field
GT: Book
Pred: Book, Magazine, Newspaper
GT: Microscope
Pred: Microscope, Monocular, Lab bench
GT: Track
Pred: Track, Barrier, Guardrails

GT: Fruit
Pred: Fruit, Banana, Broccoli
GT: Frisbee
Pred: Frisbee, Grass field, Grass
GT: Horse
Pred: Horse, Horse jump, Person
GT: Cake
Pred: Cake, Cake server, Candle

GT: Screen
Pred: Screen, Television, Window
GT: Tire
Pred: Tire, Motorcycle, Engine
GT: Computer
Pred: Computer, Desk, Screen
GT: Computer
Pred: Computer, Refrigerator, Voting booth

GT: Skateboard
Pred: Skate ramp, Skateboard, Pavement
GT: Tree
Pred: Tree log, Wood, Wood tie
GT: Vacuum
Pred: Broom, Vacuum, Toilet brush
GT: Stall
Pred: Market stand, Storefront, Stall
GT: Bowl
Pred: Person, Bowl, Bottle

Figure 7: Example predictions on GazeFollow. Annotations are shown in white and predicted points in blue. We also provide the 3 top predicted class labels, and the corresponding ground-truth label below each image. [Top] correct predictions, [Bottom] failures cases.

decides to predict the class object corresponding to the second peak, while the baseline inevitably predicts the class of the object matching the higher peak. For example, the rightmost image shows a heatmap that mainly highlights the easel while a second (and weaker) peak points to the elephant. Our model correctly predicts *Elephant* while the baseline incorrectly outputs *Easel*.

Since the baseline's recognition performance is slightly better than our model, we analyzed a set of samples where the two models didn't agree on a gaze target class. In Figure 8 (bottom), we show several samples from GazeFollow where the baseline matches the ground-truth while our model is not, despite the latter producing a reasonable prediction. For example, since *Ball* and *Balloon* are both included in the vocabulary, the model might be predicting one when the ground-truth corresponds to the other. This ambiguity is introduced by the semantic overlap in certain classes of the vocabulary (*e.g. Wave* and *Water*).

## A.3   Experimental Protocol

**Datasets.** We use both the GazeHOI and GazeFollow [42] datasets. GazeFollow is an image-based dataset annotated with head boxes and 2D gaze points. It has around $130K$ annotated instances in $122K$ images. The test set comprises $4782$ gaze instances, each of which is labeled by multiple annotators.

**Metrics.** For GazeFollow, we report the standard distance metrics (*i.e.* min and avg) following previous protocol [10] to assess gaze target localization, and we propose Accuracy@1 and Accuracy@3

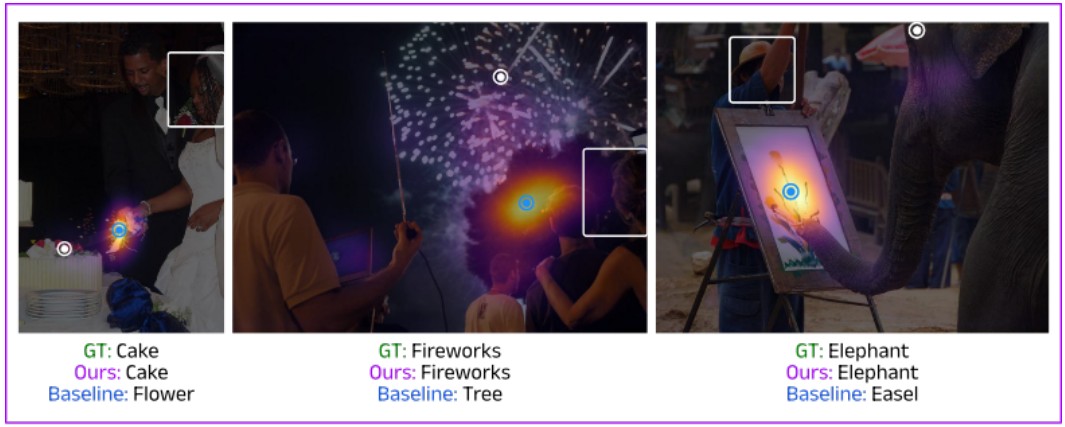

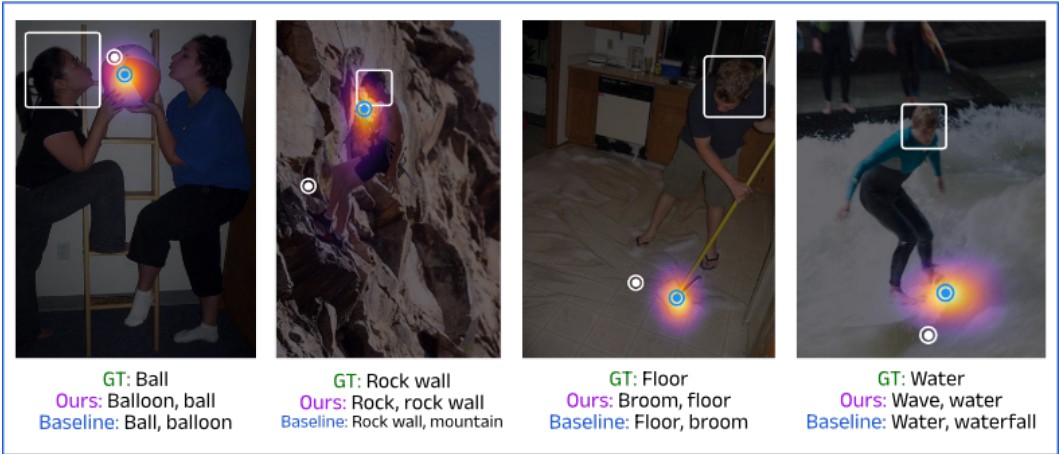

Figure 8: Comparison with the baseline. We show the annotated person in white, the predicted point in blue, and the predicted heatmap overlaid. [Top] cases where gaze location is incorrect, so the baseline's predicted class is also incorrect, but consistent with the location. However, our model's predicted label is correct. Notice the bimodal heatmap on the image with the elephant. [Bottom] Ambiguous cases where the baseline is correct, but our model is incorrect despite being semantically accurate.

for gaze target recognition. Additionally, we introduce a MultiAccuracy@1 score to account for the multiple gaze labels that can be assigned to the same instance. This metric checks if the top predicted label belongs to the set of ground-truth labels. For GazeHOI, we report a gaze accuracy instead of the distance since we don't have access to a ground-truth point. This metric checks whether the predicted gaze point falls within the ground-truth bounding box of the object. In terms of recognition performance, we use the same Accuracy@1 and Accuracy@3 as before.

**Implementation Details.** Our architecture processes the input scene image at a resolution of $256 \times 256$, and the head crop at $224 \times 224$, to produce an output heatmap of $64 \times 64$ and a label embedding of $512$. The dimension $d$ inside the decoder is set to $96$. The ground-truth heatmap uses a gaussian of $\sigma = 3$ placed around the gaze point. We use CLIP as the text encoder and keep it frozen. We set the number of blocks in the gaze decoder to 2, and the number of people during training to $N_p = 1$. The backbone in the gaze encoder is a ResNet-18 pretrained on Gaze360 [28], while the image encoder is a ViT-base model [12] initialized from a multimodal MAE [3]. Finally, our experiments on GazeHOI are initialized from a model trained on GazeFollow.

**Training.** For the main experiments on GazeFollow, we use the AdamW optimizer with a learning rate of $2e - 4$ and weight decay of $0.003$. We train for 20 epochs , with a warmup of 4 epochs, and a cosine annealing schedule. The batch size is set to 300, and the loss coefficients for $\lambda_{hm}$, $\lambda_{lab}$, and $\lambda_{ang}$, are set to 1000, 1, and 3 respectively. We also use stochastic weight averaging [24] for better generalization. On GazeHOI, the only change is to remove the warmup, and reduce the learning rate

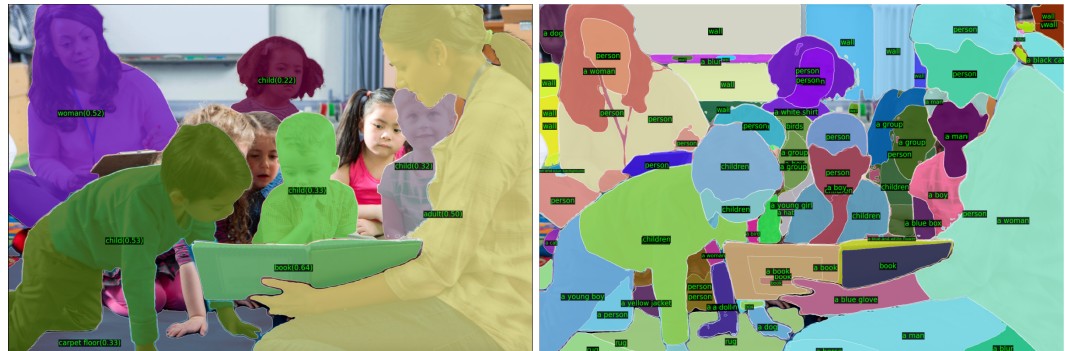

Figure 9: Comparison of the outputs of the segmentation methods. Left: RAM-Grounded SAM. Right: Semantic-Segment-Anything.

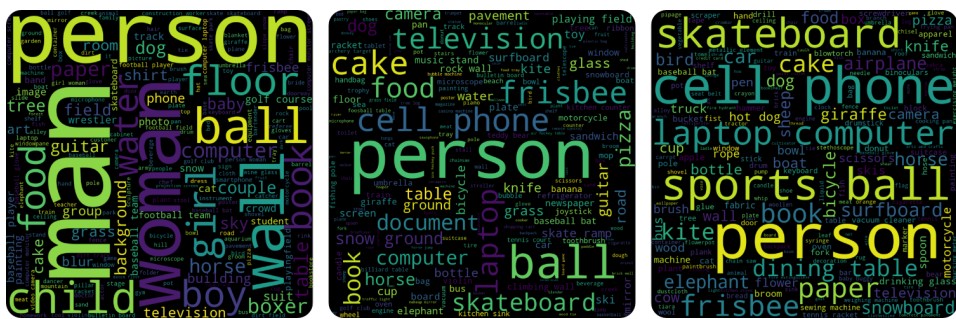

Figure 10: Word clouds of the vocabulary of gaze labels. Left: pseudo-labels of GazeFollow. Middle: ground-truth labels of the test set of GazeFollow. Right: ground-truth labels of GazeHOI.

to $2e - 5$. All experiments are done on either a single RTX 3090 (24Gb of memory) or H100 (80Gb of memory) depending on memory requirements, and last for 2 to 10 hours each.

When training on GazeHOI, we use the center point of the object's box as a proxy for the gaze target point. However, for larger objects this strategy is not optimal. For instance, the center point of a person's box will be around the waist area while the actual gaze point will typically be located on the head (*e.g.* bottom right sample of figure 2). To overcome this issue, when object boxes are larger than a threshold, we use a prediction from a model trained on GazeFollow as the ground-truth point as long as it falls within the object's ground-truth box, otherwise, we default back to the center point.

**Validation.** GazeFollow doesn't propose any validation split, we use the train/val splits proposed in [48]. The best models on the validation set are selected based on the distance score for GazeFollow and the validation loss for GazeHOI.

**Inference.** To perform inference with our models, we first build a vocabulary of classes (or use one from the datasets), pass them through the text encoder to obtain the text embeddings, and discard the text encoder. The label predictions can now be made by comparing the predicted gaze label embedding to the text embeddings and selecting the top-k based on the similarity score.

**Coordinate Decoding.** Vision tasks where the goal is to predict pixel coordinates on an image are typically formulated as a heatmap prediction (*e.g.* facial keypoints detection, pose estimation, gaze following). For computational reasons, the heatmaps are often set to a much lower resolution than the input. This introduces quantization errors when considering the $\arg\max$ of the heatmap as the predicted point. In this paper, we use the DARK [52] coordinate decoding scheme, which accounts for the distribution of the heatmap when estimating the continuous coordinate values.

**Extra People.** Generally speaking, both datasets annotate a single person per image, even when there are more. In the experiments where we need to train with more than one person, we use an off-the-shelf head detector to find other *extra* people. Since they don't have associated gaze points, we only backpropagate the loss from the person that is annotated. We use padding or truncation of people's heads and box coordinates as necessary to ensure that $N_p$ is fixed during training.

## A.4 Annotation Protocols

### A.4.1 Label Generation

In Figure 9, we provide a comparison of the outputs of the open-world segmentation methods used to derive the pseudo-labels for GazeFollow. Please note that in case the pixel corresponding to the annotated gaze point isn't segmented by any of the two methods, we simply use the closest segmented pixel from the second approach (*i.e.* Semantic-Segment-Anything).

In Figure 10, we provide a word cloud of the vocabulary frequencies of the pseudo-labels of GazeFollow (left), the ground-truth labels of the test set of GazeFollow (middle), and the ground-truth labels of GazeHOI (right).

### A.4.2 Annotation Teams

The test set of GazeFollow was annotated by a single person to ensure consistency of class labels. For GazeHOI, the manual verification step was done by 3 people.

| Experiment | Avg. D. ↓ | Min. D. ↓ | Acc@1 ↑ | Acc@3 ↑ |
|---|---|---|---|---|
| CLIP Vision | 0.148 | 0.087 | 0.341 | 0.485 |
| Supervised | 0.129 | 0.069 | 0.437 | 0.628 |
| MAE | 0.112 | 0.056 | 0.442 | 0.621 |
| DinoV2 | 0.113 | 0.055 | 0.465 | 0.658 |
| MultiMAE | 0.108 | 0.051 | 0.450 | 0.636 |
| Random | 0.107 | 0.052 | 0.459 | 0.630 |
| BERT | 0.109 | 0.052 | 0.449 | 0.632 |
| MPNet | 0.110 | 0.054 | 0.422 | 0.615 |
| CLIP Text (FT Proj) | 0.108 | 0.052 | 0.439 | 0.633 |
| CLIP Text (Frozen) | 0.108 | 0.051 | 0.447 | 0.642 |
| $N = 1$ | 0.109 | 0.053 | 0.444 | 0.647 |
| $N = 2$ | 0.108 | 0.051 | 0.447 | 0.642 |
| $N = 4$ | 0.107 | 0.051 | 0.447 | 0.634 |

Table 4: Ablation results on GazeFollow for the image encoder's pretraining, the text encoder's pretraining, and the number of decoder blocks (in order).

## A.5 More Ablations

**Vision Pretraining.** In this section, we assess the influence of the pretrained weights of the image encoder on the final performance of the model. To this end, aside from our MultiMAE weights, we try CLIP's pretrained vision encoder, a ViT trained on ImageNet, standard MAE [23], and DinoV2 [38]. Please note that Dino uses an image resolution of $224 \times 224$ and a patch size of $14$ instead of resolution of $256 \times 256$ and a patch size of $16$. However, the number of output tokens remains unchanged. Also, for memory restrictions, we use a batch size of $216$ instead of $300$. The results are shown in Table 4 (top row). First, we note that using CLIP's vision weights to perform both localization and recognition degrades performance significantly (*i.e.* $0.148$ *vs.* $0.108$ Avg Dist). This serves to prove that the design of our baseline required CLIP's vision part to only perform the recognition task, not localization. Aside from CLIP, both MAE and Dino offer performance that is similar to our MultiMAE. In other words, our architecture is not very sensitive to the pretraining of the image encoder.

**Text Pretraining.** We also perform an ablation to evaluate the influence of the text embeddings on the final performance (*cf*. Table 4, second row). For this experiment, we try random projections, BERT [11], MPNet [46], CLIP's text encoder where we fine-tune the final projection layer, and our version with the frozen CLIP. The performance for both localization and recognition seems unaffected by the type of text encoding used for the vocabulary. Surprisingly, even random projections seem to perform on par with the rest of the methods. However, random projections severely limit the generalization ability of the model as they can only work on the dataset and labels they were trained

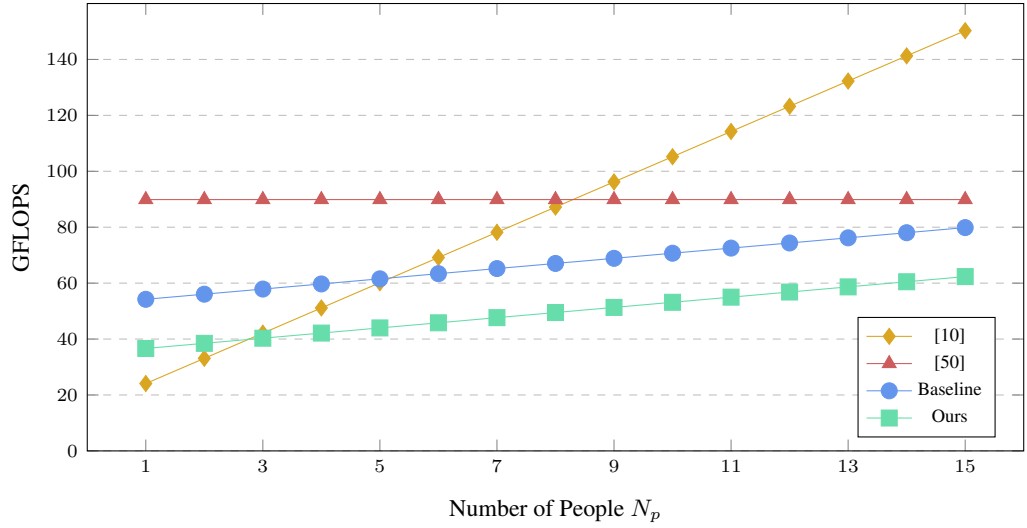

Figure 11: Comparative analysis of FLOPS *vs* number of people.

on. To demonstrate this, we perform a cross-dataset evaluation on GazeHOI with these random projections as class embeddings. We obtain a similar localization performance for both methods, but we get an Acc@1 of $26\%$ and Acc@3 of $39\%$ compared to our frozen CLIP text variant which obtains $31\%$ and $46\%$, respectively (these are the same values reported in Table 2 of the main paper).

**Gaze Decoder.** The final ablation is on the number of blocks used in the transformer of the gaze decoder. From Table 4 (third row), we can see that performance seems to be optimal for $N = 2$, although the differences are not significant.

### A.6 Model Efficiency

We mentioned previously that the baseline has a larger parameter count than our proposed model. In this section, we provide a FLOPS comparison as a function of the number of people in the image to illustrate the efficiency and graceful scaling of our architecture's design. Aside from our model and baseline, we also show [10], which is a popular two-stage method like ours, and [50] which is a single-stage model (*i.e.* it simultaneously predicts all people's heads and gaze heatmaps at the same time, instead of using a separate head detector). To ensure a fair comparison with [50], we include the cost of head detection in all other methods. Please note that we use the RGB-only variant of [50], instead of the RGB+Depth variant (*i.e.* the best performing one). Using the latter would increase the cost even more to account for depth extraction.

As we can see in Figure 11, [50] is constant as expected, while [10] shoots up since the early fusion of the architecture means that the expensive scene encoding is repeated for each person. On the other hand, our model displays a more modest increase due to the lightweight decoder, and the scene encoding being executed only once. We also observe that the baseline surpasses our model by about 20 GFLOPS due to the extra ViT (which is also ran once per image). It is worth noting that even at $N_p = 15$, our model is still largely more efficient than the constant one-stage [50].

