# OpenReview forum: "Toward Semantic Gaze Target Detection"
_NeurIPS.cc/2024/Conference — NeurIPS 2024 poster_

### Official Review · Reviewer_kgqK · 2024-07-11

**Soundness:** 3
**Presentation:** 3
**Contribution:** 3
**Rating:** 4
**Confidence:** 5

**Summary:**

This paper proposes simultaneously detecting a person's gaze location and recognizing the class label of the gaze target.  To complete the task, a new architecture is designed. The architecture is efficient when multiple people are present in the image because it processes the scene image once and detects each person's gaze by decoding the correlations between their head information and the scene image tokens. To evaluate the method, this paper annotates the target label in the existing GazeFollow dataset and collects GazeHOI based on several existing datasets. On benchmark GazeFollow and GazeHOI, the proposed architecture performs better in localizing the gaze target than the existing gaze-following method.

**Strengths:**

1.  This paper simultaneously detects the location and semantic label of a person's gaze target in a scene image. The task is meaningful and has been explored rarely.
2.  Benchmark is collected with both gaze target location and semantic label.
3.  The component of gaze decoder is interesting. The proposed method is efficient when multiple people are present in the image because it processes the scene image once and detects each person's gaze by decoding the correlations between their head information and the scene image tokens.  It avoids the computation burden in previous methods that merge the scene images and head images before the image encoder.
4. The proposed method outperforms other method in localizing the gaze target.

**Weaknesses:**

1. According to the experimental results (Tab 1&2), the proposed method does not show advantages in recognizing the label of gaze target rather than the baselines. The baselines also show equivalent performance in localizing the gaze target. Therefore, the baseline (heatmap) is shown to be the best solution in simutaneously detecting the location and semantic label of a person's gaze target.

2. The introducing of L_lab in the proposed method is not necessary. First, adding L_lab does not improves the ability the gaze target localization experimentally (Tab. 3). Second, L_lab mechanism enpowers the architecture to recognize the semantic label of gaze target, but it could be designed in a simpler and more effective way, as shown by the baselines in Tables 1 and 2.

3. Details of the gaze decoder (the third and fourth paragraph in Sec 3.3) is not presented clearly. It could be clearer if the intermediate x_gaze and x_img are marked in Fig. 1.

**Questions:**

Please refer to the weakness.

**Limitations:**

The authors have adequately addressed the limitations.

---

> ### Author Rebuttal · Authors · 2024-08-06
>
> **Necessity of $L_{lab}$. It doesn’t improve localization, and the semantics can be predicted in a simpler and more effective way by the baseline**
>
> The $L_{lab}$ loss is necessary for joint training (cf. our reply in the *overall rebuttal* on the motivation for joint training). It is not meant to improve localization. However, even if it is not reflected on the current datasets/metrics, we believe it could improve localization since the task of selecting what a person looks at in a given context after a field of view has been determined, is inherently a semantic problem.
>
> Regarding the comparison, we fail to see how adding and fine-tuning an entire ViT (86.6M params) is “simpler” than an extra MLP head (3.3M params). Please check our reply to reviewer **9bSU** about efficiency. As for effectiveness, we refer the reviewer to our answer on the comparison with the baseline to better interpret the numbers (cf. *overall rebuttal*).
>
> **More explanation of the decoder**
>
> We wish to thank the reviewer for recognizing the value of our decoder’s design. We will take steps to improve the description and annotate the main figure with $x_{gaze}$ and $x_{img}$ for better readability. Below, we further explain paragraphs 3 and 4 of section 3.3 as requested.
>
> At first, $x_{img}$ (which represents image tokens) and $x_{gaze}$ (which represents gaze tokens, one per person) go through a set of transformer blocks composed of a series of cross-attention and feed-forward operations combined with residual connections. These cross-attention operations go in two ways, one where the gaze tokens $x_{gaze}$ generate the queries, while the image tokens $x_{img}$ generate the keys and values (we call this $x_{gaze}$ → $x_{img}$ or gaze to image cross-attention), and one where image tokens generate the queries and gaze tokens generate the keys and values (we call this $x_{img}$ → $x_{gaze}$ or image to gaze cross-attention). The two-way design allows image tokens to incorporate information from gaze tokens, and vice-versa. This helps them better align for the final dot-product operation that will produce the final heatmap.
>
> On Figure 1 of the paper, $x_{img}$ flows through the decoder following the blue arrows until the *Upscaler* module, after which it becomes $\hat{x}_{img}$ (L138-139), which is simply a spatially upscaled version of the image feature map representation with a lower number of channels.
>
> On the other hand $x_{gaze}$ flows through the decoder by taking the path denoted by the orange arrows until the two output MLP heads. The bottom MLP predicts the gaze label embedding (one per person), while the top one produces $\hat{x}_{gaze}$ (L136-137).
>
> To obtain the final heatmap, we simply perform a dot-product between $\hat{x}_{img}$
>
> and $\hat{x}_{gaze}$ (L141-144).
>
> Finally, the *not gaze token* is simply an extra learnable token, much like the CLS token in standard transformers. However, unlike the CLS token which aggregates information due to the final supervision, our extra token is not supervised and plays a different role. Conceptually, it serves to improve the image to gaze cross-attention. Without this token, this operation will update each image token with a weighted sum of the value embeddings derived from gaze tokens. But what if nobody is looking at the area represented by that image patch? We need to allow the model to flexibly choose a different “background” token in that weighted sum, instead of forcing only information from those people’s gaze tokens to update that image patch. The *not gaze token* plays the role of this “background” placeholder, a hypothesis that we verify in Figure 4 (last row) and L331-340.
>
> **Comparison with the baseline**
>
> Please check our detailed reply to this comment in the *overall rebuttal* section.

---

### Official Review · Reviewer_9bSU · 2024-07-12

**Soundness:** 3
**Presentation:** 3
**Contribution:** 2
**Rating:** 5
**Confidence:** 4

**Summary:**

This paper introduces a novel approach to semantic gaze target detection, extending the traditional gaze following task to include not just localization of where a person is looking, but also identification of what they are looking at. The authors address the limitation of existing gaze following methods that only predict pixel coordinates, proposing to also identify the semantic label of the gaze target. They introduce an end-to-end architecture that simultaneously predicts both the location and class label of the gaze target, where the recognition part is framed as a visual-text alignment task. They also create new benchmark datasets through pseudo-annotation of existing datasets. The proposed method demonstrated superior performance, especially in the localization accuracy of the gaze target, surpassing existing methods.

**Strengths:**

- The paper is written clearly overall, and the motivation is also evident.
- The method design is also clear, and it is interesting that this approach improves localization accuracy.

**Weaknesses:**

- While interesting, it is not entirely clear why training semantic label prediction simultaneously improves localization accuracy, and this point requires further detailed discussion. Whether it is appropriate to simply interpret that the simultaneous training approach has achieved state-of-the-art performance still needs to be investigated.
- The computational efficiency benefits of simultaneous estimation are understandable, but there are few experimental details reported on this point. The abstract mentions "with 40% less computation," but it was unclear how this number was derived.

**Questions:**

- Please clearly explain the advantages of simultaneously estimating semantics (whether the improvement in accuracy truly stems from this problem setting).
- If there are any experimental evidence on computational efficiency within the paper or appendix, please indicate them.

**Limitations:**

It appears that a reasonable argument is presented in the paper.

---

> ### Author Rebuttal · Authors · 2024-08-06
>
> **More experimental evidence on computational efficiency. How did we get the 40% decrease in parameters?**
>
> In terms of parameter count, our model features 116M while the baseline has 200M. This is because our model only uses an MLP head (3.3M params) for label prediction while the baseline uses a ViT encoder (86.6M params). That’s a decrease of 42% in the total number of parameters.
>
> To be more thorough, we also provide a FLOPS comparison as a function of the number of people in the image to illustrate the efficiency and graceful scaling of our architecture’s design (cf. Figure 1 left panel of the PDF). Aside from our model and baseline, we also show Chong et al. [10] which is a popular two-stage method like ours, and Tonini et al. [46] which is a single-stage model (ie. it simultaneously predicts all people’s heads and gaze heatmaps at the same time, instead of using a separate head detector). To ensure a fair comparison with [46], we include the cost of head detection in all other methods. Please note that we use the RGB-only variant of [46], instead of the RGB+Depth variant (i.e. the best performing one). Using the latter would increase the cost of [46] even more to account for depth extraction.
>
> As we can see from the graph, [46] is constant as expected, while [10] shoots up since the early fusion of the architecture means that the expensive scene encoding is repeated for each person. On the other hand, our model displays a more modest increase due to the lightweight decoder and the scene encoding being executed only once. We also observe that the baseline surpasses our model by about 20 GFLOPS due to the extra ViT (which is also ran once per image). It is worth noting that even at 15 people, our model is still largely more efficient than the constant one-stage [46].
>
> We will add a section specifically dedicated to the efficiency dimension to the final version of the paper.
>
> **Motivation for joint detection of heatmap and object class. Does joint training improve localization?**
>
> We refer the reviewer to our detailed reply to this comment in the *overall rebuttal* section.

---

### Official Review · Reviewer_DJCe · 2024-07-12

**Soundness:** 3
**Presentation:** 4
**Contribution:** 3
**Rating:** 7
**Confidence:** 3

**Summary:**

In this work, the authors propose an enhancement to classical gaze-following prediction, which typically operates on a 2D coordinate level, by integrating semantic label prediction. To this end, they introduce a novel architecture inspired by promotable segmentation, incorporating a multi-input transformer. This architecture uniquely combines scene images and head-related attributes (bounding coordinates and head crops) through a disjoint fusion process. This allows the gaze encoder to process multiple user pairs without additional processing load, making the model uniquely applicable to a wide range of scenarios.

Furthermore, using the GazeFollow dataset, the authors present a robust pseudo-annotation pipeline and introduce a new benchmark for the community's benefit.

The paper is well-written, with a coherent narrative that provides ample evidence of the method's effectiveness through a series of ablation studies.

**Strengths:**

1. Extending the existing gaze-following task with semantic cues enhances model explainability by introducing a semantic gaze-following task. Instead of merely predicting 2D coordinates on an RGB image, the authors incorporate semantic components, significantly improving overall performance.

2. The introduction of the label loss function involves calculating the cross-entropy of the cosine similarity between predicted and true visual gaze label embeddings. This label loss follows the contrastive InfoNCE formulation, a key component in self-supervised models that learn meaningful representations from negative and positive pairs. The analogy is applied here by comparing positive and negative segmentation labels.

3. The combination loss includes an angular loss, which provides additional interpretability for the predictions.

4. A supportive benchmark is proposed to evaluate model performance in an unbiased manner, allowing the community to build upon it.

5. Empirical evaluations confirm the model's performance, demonstrating state-of-the-art results on the main GazeFollow task. The provided ablation study helps readers understand the contribution of individual components, such as the loss function.

**Weaknesses:**

For the proposed baseline, where the pipeline is robust, I could consider several additional data augmentation techniques, such as flipping and distorting image patches. This could further enhance the pipeline.

According to the supplementary material, the model performs poorly in scenes with specific angles and relationships between the subject and object. To further improve overall performance, this problem should be considered in a 3D space, requiring a different dataset with fully labeled scenes. This approach would help mitigate partial occlusions by incorporating depth information.

A dataset is a crucial component of this work. Therefore, I suggest including some statistical comparisons to thoroughly track the pseudo changes in the GazeFollow dataset. Currently, the approach seems to be at the level of omitting synonyms, among other methods. However, I assume this was not entirely feasible.

**Questions:**

A typo in line 161, where instead of constrastive, please fix it to contrastive.

Do I understand correctly, that semantic label embeddings have the form of natural language words,e.g. cup, bad, sock, etc or some other form of encoding/transformation was adopted here? It is relevant for the label loss function.

Is it planned to release the source code?

**Limitations:**

Those provided failure cases are indeed very relevant, therefore, I recommend additionally considering additional forms to represent data like 3d to omit those failures.

---

> ### Author Rebuttal · Authors · 2024-08-06
>
> We are grateful to the reviewer for the valuable suggestions. Here are our comments:
>
> **Data augmentation for the baseline**
>
> We agree with the reviewer. The baseline variants that are fine-tuned already use data augmentation techniques like flipping and color jittering.
>
> **Use of depth information to model the 3D scene**
>
> This is a great idea. In fact, the earlier versions of our architecture used depth information (extracted from the RGB image) to reconstruct the 3D scene point cloud and infer a 3D visual field of the person from a predicted 3D gaze direction. This field map was used to filter out image areas where the person could not be looking (in 3D) given their head and eye direction. As expected, this addition improved localization performance. Ultimately, we decided to drop this component for 2 reasons:
> 1. Using depth to improve performance is already well documented in the literature [5, 16, 20, 26, 45, 46]
> 2. Between the new task, the datasets, the evaluation protocols, the baselines, and the new architecture, our paper contains numerous contributions as it is, and we deemed it preferable to keep the focus on the topic of semantic gaze following without adding unnecessary complexities.
>
> **More statistics on GazeFollow’s pseudo-labels and cleaning steps**
>
> We appreciate the suggestion. We will expand the section on GazeFollow’s pseudo-annotations to provide more details in the final version of the paper. The released annotations will also include the original pseudo-labels from both segmentation methods used in the paper in case other researchers want to process or combine them differently.
>
> **Clarification on semantic label embeddings**
>
> The semantic label embedding of the gaze target (or simply gaze label embedding) is a vector of size 512 that captures semantic information about the visual area the person is looking at. This embedding is trained to match the text embedding (produced by the text encoder) of the ground-truth class via a contrastive loss. At test time, we use a vocabulary of classes (e.g. person, tree, table) that is converted into class embeddings by the text encoder. Then, given a predicted gaze label embedding, we find the closest class embedding (using cosine similarity) to get the predicted object class (or gaze label) from the vocabulary.
>
> **Release of source code**
>
> Of course! The source code, datasets and model checkpoints will be made publicly available upon acceptance. Our hope is that this will attract more researchers to work on this topic, and encourage the community to build upon our work.
>
> **Typo**
>
> Thank you for flagging the typo, we have corrected it.

---

> > ### Comment · Reviewer_DJCe · 2024-08-13
> > **Thank you for your response**
> >
> > Thank you for your response. My positive evaluation of the paper also remains after reading reviews and rebuttals.

---

> ### Author Response · Authors · 2024-08-13
>
> Dear Reviewer DJCe,
>
> Thank you once again for the positive evaluation. We appreciate your feedback and will be updating our paper with your suggestions.
>
>
> Best.
>
> Authors.

---

### Official Review · Reviewer_roY1 · 2024-07-16

**Soundness:** 2
**Presentation:** 3
**Contribution:** 2
**Rating:** 5
**Confidence:** 4

**Summary:**

The authors propose an architecture capable of predicting both the gaze target location and the semantic class of a person's gaze target in an image, representing an advancement over traditional methods that only predict the pixel coordinates of gaze fixations. They introduce new benchmark datasets and experimental protocols, leveraging tasks from gaze following and human-object interaction. The task of gaze target detection is particularly valuable for human-centric AI.

**Strengths:**

1.The integration of semantic analysis into gaze following is a significant step forward.
2.The development of benchmarks provides a foundation for future research, assuming they are publicly released.

**Weaknesses:**

1.Compared to "Object-aware gaze target detection" from ICCV 2023, the improvements and extensions are limited. Although the model outputs both a gaze heatmap and its class, the paper fails to convincingly justify the necessity for detecting the object class, especially when a conventional object detection or classification model could achieve similar results after obtaining the heatmap.
2.The category labels for the GazeFollow training set and the head crops for GazeHOI are generated using existing models. This reliance on pseudo-labels potentially limits the data's value, despite manual re-checks for the head crops.
3.The paper primarily showcases examples from the GazeHOI dataset, with few examples and details from the GazeFollow dataset.
4.The proposed model does not outperform the baseline in localization and is surpassed by the Baseline (heatmap weight) in recognition, as demonstrated in Tables 1 and 2. This suggests that the sophisticated model structure presented in Section 3 has limited effectiveness.
5. The references have numerous formatting errors, such as those in [1, 46].

**Questions:**

1. Please address the concerns raised in the Weaknesses
2.Additionally, will the proposed datasets and models be made publicly released soon?

**Limitations:**

The authors have indeed addressed the limitations of their work to some extent in the manuscript.

---

> ### Author Rebuttal · Authors · 2024-08-06
>
> **Limited improvement compared to Tonini et al. (2023) [46]**
>
> We respectfully disagree with the reviewer. Here are the significant differences
> - [46] does not predict a gaze target category (L87-89), it merely uses general object detection as an auxiliary task to improve gaze target localization
> - [46] is limited to the 80 categories of COCO whereas our weakly-supervised training learns about significantly more semantic concepts (eg. 463 in GazeHOI)
> - Learning general object detection alongside gaze following (which are two mostly unrelated tasks) using the same backbone is difficult, and significantly penalizes the accuracy of both. In fact, in most cases, [46] can’t even detect people’s heads, let alone predict their gaze. We provide several samples from GazeFollow’s test set in Figure 1 (right panel) of the PDF where [46] can’t properly localize heads, even easy ones (the predicted gaze locations seems random or based on statistical priors). The GT person in these samples is not detected at all, which raises questions about what their reported performance values actually mean. In fact, we ran [46] on 200 images from the test set of GazeFollow, and found 15% of cases where the GT person’s head is not overlapping with any predicted box, and this is not including other issues.
> - The architecture design is very different: [46] uses a DETR and passes the embeddings of detected persons and objects to a standard transformer decoder. Our model follows a more principled approach by framing the task as promptable gaze following, where a scene encoder highlights gaze candidate regions and a novel decoder (recognized as interesting by reviewer **KgpK**) decodes the heatmap and gaze embedding jointly by aligning representations. The insights on scene encoding and gaze decoding are supported by evidence (L313-329 and L331-340) and make the architecture conceptually more intuitive, and what the model is learning easier to explain (not to mention the largely better localization performance).
> - Our model is significantly more efficient (cf. Figure 1, left panel in the PDF)
> - [46] proposed an incremental architecture for an already established task. **Our paper is proposing a new task, new datasets and annotations, a novel architecture achieving SOTA localization (outperforming previous SOTA by a large margin) and strong recognition, evaluation protocols and performance metrics.** The positioning of the two papers is not the same, we kindly ask the reviewer to keep this in mind.
>
> **Necessity of detecting the gaze target class**
>
> As stated in L36-37, most gaze following applications do not particularly care about the pixel location where a person is looking, but rather the underlying semantics. For example, in autism screening, clinicians mostly look for specific gaze behavioral patterns that can not be identified without the semantic class (e.g. eye-contact or alternating between looking at a toy and the clinician). The semantics can also enable context-aware responses. For example, if a person is looking at a potentially dangerous object, the system can trigger appropriate alerts or actions.
>
> From a technical perspective, predicting both location and class can help disambiguate the intent, especially in scenes where multiple objects are in the field of view of the person. Also, since the area covered by the heatmap often includes multiple objects, the semantic class can help select a target based on more than just visual saliency, and refine its location.
>
> The importance of the semantic component was also argued in [45], where authors propose a metric to assess models semantically, stating that localization metrics alone can be misleading (ie. even smaller distances can translate to different objects while larger distances can reflect the same object if it’s big enough).
>
> **Reliance on pseudo-labels potentially limits the data’s value**
>
> A) Regarding the class labels of the training set of GazeFollow, we agree that pseudo-labeling is not perfect, however, please keep in mind that:
> - Annotating a gaze class from scratch is very difficult as explained in L206-209
> - Many papers today leverage pseudo-annotation and weakly-supervised training with great success (e.g. [37])
> - The large vocabulary of pseudo-labels makes it possible for models trained on GazeFollow to acquire a rich semantic understanding, and generalize to other semantically close vocabularies. For example, we can distinguish between “man”, “woman”, and “child” instead of the generic “person” category found in most vision-based vocabularies. Also, note in Table 2 that the cross-dataset evaluation (ie. no fine-tuning) on the vocabulary of GazeHOI, shows comparable performance to the zero-shot baselines relying on the large-scale semantic pre-training of CLIP.
>
> B) As for the head boxes of GazeHOI, the statement that they are pseudo-annotations is simply not true. A model-based annotation that is verified by a human annotator is, in fact, a ground-truth (and the default approach nowadays, e.g. [28]). It is perhaps worth noting that GazeHOI initially contained about 100k images, more than 50% of which were discarded during that manual verification step. Furthermore, from our past experience with this specific case, we can confidently say that a model-assisted approach works better than human annotation because the pre-trained head detector we use is extremely accurate and tightly delimits the heads. A human annotation from scratch at this scale often results in looser and inaccurate bounding boxes as can be observed in GazeFollow’s original annotations.
>
> **Qualitative samples from the GazeFollow dataset**
>
> We have focused on GazeHOI because it was an entirely new dataset. For the sake of completeness, we will also add an extra page of qualitative samples (annotations, good predictions, and failure cases) from GazeFollow to the final version of the paper. In the meantime, we provide many samples in Figure 2 of the attached PDF for reference.

---

> ### Comment · Reviewer_roY1 · 2024-08-12
> **My concerns have been well addressed**
>
> Thanks for the response. I have carefully read the comments and other reviews. My concerns have been addressed and I lean towards changing my rate to borderline acceptance.

---

> ### Author Response · Authors · 2024-08-12
> **Thank you**
>
> Dear Reviewer roY1,
>
> We would like to express our gratitude for your thoughtful review and for taking the time to consider our rebuttal. We are happy to know our reply addressed all your concerns, and we appreciate you increasing your rating. We will incorporate your invaluable feedback and subsequent discussion into the revised version of the paper.
>
> Best.

---

### Author Rebuttal · Authors · 2024-08-06

We extend our gratitude to the reviewers for their thoughtful feedback. We address below the common concerns, and will incorporate the discussion in our final version.

As a reminder, the goal of this paper is to establish the foundation for the novel, significant and challenging task of semantic gaze following by introducing benchmarks, evaluation protocols, and model architectures achieving SOTA for gaze following.

**roY1, 9bSU    Motivation for joint detection of heatmap and object class. Why not use a separate object detector? Does joint training improve localization?**

We have motivated the joint training in L38-54 using efficiency (which **9bSU** acknowledged) as our main argument. Given the importance of this question, perhaps we should elaborate more:
- First, combining localization and recognition is the more natural formulation. Otherwise, why not also decouple object detection into object localization and image classification with two networks?
- There is no object detector or classifier that can recognize all classes of our annotations. Plus, we also have to deal with uncountable objects (ie. Stuff classes).
- Joint training means doing both tasks using the same backbone, which is preferable when they are tightly related like in our case. Decoupling means adding an entire separate network instead of our MLP head, which is unnecessarily inefficient.
- Unlike object detection, in gaze following we predict heatmaps and not boxes, which makes applying a separate object detector afterwards more challenging than **roY1** implies. How do you match a heatmap to the right box? Should you consider the argmax of the heatmap as the location of the object and match it with the boxes? What if the point falls within multiple boxes? Or instead, should you consider the energy of the heatmap contained in each box? How do you deal with the different scales of overlapping boxes then? Also, what if the heatmap is multimodal, how do you make your selection? We would need to craft rules and heuristics tailored to each use-case, and will almost inevitably be suboptimal. **Joint training allows the model to learn the best way to dynamically make sense of that heatmap in order to infer the right class.** For example, we found several instances where the predicted gaze label is correct despite the location being incorrect. This can happen when the heatmap is multimodal and the model decides to select the class of the second peak instead. In this case, using a separate model to match the predicted argmax location will surely lead to the wrong class (cf. Figure 3 left panel of the PDF).

Finally, we achieve SOTA in gaze following localization performance due to our novel architecture (promptable gaze following formulation and decoder design). We do not claim in our paper that joint training improves localization. In fact, it doesn’t, at least not on our datasets. Instead, we argue that it’s difficult to do the extended task any other way that is as efficient, equally performant, and more natural. That being said, joint training definitely improves recognition (+3.4% flat points on Acc@1, cf. Table 3).

**roY1, kgqK    Comparison with the baseline. Localization is the same as the baseline, and recognition is slightly worse**

There seems to be a misunderstanding. The baseline uses a two-step process: first, a frozen gaze model predicts a gaze heatmap; then, a second network performs class recognition using the original image fused with the heatmap.

In our experiments, **we use our own novel gaze model to generate the heatmap for the baseline, so the localization performance is the same by design** (L234-237). This allows us to control for the localization factor, enabling an unbiased comparison of recognition performance (L242-244).

As for recognition accuracy, while the heatmap variant is slightly better, it is important to consider that the design of our baselines is unfair to our model:
- All baselines use our own model’s localization, which is the current SOTA by a fair margin (L261-262). Thus, the baseline literally needs our model to achieve that recognition accuracy. To verify this, we swap our gaze model with the one from [10] and find that Acc@1 and Acc@3 drop from 0.466 and 0.653 to 0.442 and 0.620 respectively on GazeFollow. Under this setting, the baseline becomes worse than our model.
- The baseline uses an extra CLIP encoder instead of our MLP head, making it computationally more expensive (L262). The heatmap variant is also fine-tuned separately, which means extra training costs. **Attempting to use the same CLIP vision encoder to do both localization and recognition significantly degrades performance** (cf. Table 4, first row)
- The baseline benefits from the large-scale (400M samples) semantic pre-training of CLIP (L262-263), whereas our recognition head is trained from scratch only on 100K samples (L275-277)
- The baseline (esp. the heatmap variant) is novel, and provides interesting insights. It was designed from scratch, and should be considered a research contribution on its own. This was recognized by reviewer **DJCe. In hindsight, maybe the term “baseline” was not the right one to use here**
- As explained in L265-268, an error analysis revealed that many cases where our method and the heatmap baseline didn’t agree were ambiguous, either due to the hierarchy or semantic similarity of the predicted and GT classes. We show a few samples in Figure 3 (right panel) of the PDF

**roY1, DJCe    Release of source code and datasets**

As stated in the checklist, we confirm that the source code, datasets and model checkpoints will be made publicly available upon acceptance. We are eager to see how the community will build upon our work.

**roY1   Reference formatting**

Thank you, they have been corrected.

We hope that our arguments provide a compelling case, and kindly ask all reviewers to contextualize the numbers and consider all contributions when assessing the merits of our paper.

---

### Author Response · Authors · 2024-08-11
**Any Last Questions?**

Dear reviewers,

We hope our rebuttal addressed your concerns and clarified any misunderstandings. As the author-reviewer discussion period will end soon, we wanted to know if you had any additional questions we can assist you with.

We are thankful for your time.

---

### Public Comment · ~Jiaren_Zhao1 · 2025-09-07
**Where is your dataset?**

Dear Samy, Anshul, Victor, and Dr. Odobez:

Thanks for your interesting work. We reviewed your paper one year ago and I am wondering when will you release the GazeHOI dataset? I remembered you admited to release it during the rebuttal but I did not see the link.

Best,
Andy

---

> ### Public Comment · ~Samy_Tafasca1 · 2025-09-08
>
> Dear Andy,
>
> The codebase and dataset have been ready for a while, we're just waiting for our lab's data manager to approve it before the formal release. It should be available soon under https://github.com/idiap/semgaze.
>
> Thank you for your patience.
>
> Best,
> Samy

---

### Decision · Program_Chairs · 2024-09-25

**Decision:**

Accept (poster)

**Comment:**

The paper introduces a two stage technique to perform both gaze localization and classification.  It also introduces a benchmark dataset.  It receives 4 reviews, including one accept, two borderline accepts , and one borderline reject .  Overall, reviewers like this work . They believe adding semantic labels to gaze following improves its explainability, and that  using pseudo labels and self-supervision loss reduce data dependence and improves its generalization.   The main concerns include marginal technical novelty and marginal performance improvement, computational complexity of the method , and  missing some  technical details .  The authors rebuttal addresses some of the concerns.